# EVALUATING ROBUSTNESS OF COOPERATIVE MARL: A MODEL-BASED APPROACH

## ABSTRACT

In recent years, a proliferation of methods were developed for cooperative multi-agent reinforcement learning (c-MARL). However, the robustness of c-MARL agents against adversarial attacks has been rarely explored. In this paper, we propose to evaluate the robustness of c-MARL agents via a model-based approach, named **c-MBA**. Our proposed formulation can craft much stronger adversarial state perturbations of c-MARL agents to lower total team rewards than existing model-free approaches. In addition, we propose the first victim-agent selection strategy and the first data-driven approach to define targeted failure states where each of them allows us to develop even stronger adversarial attack without the expert knowledge to the underlying environment. Our numerical experiments on two representative MARL benchmarks illustrate the advantage of our approach over other baselines: our model-based attack consistently outperforms other baselines in all tested environments.

## 1 INTRODUCTION

Deep neural networks are known to be vulnerable to adversarial examples, where a small and often imperceptible adversarial perturbation can easily fool the state-of-the-art deep neural network classifiers (Szegedy et al., 2013; Nguyen et al., 2015; Goodfellow et al., 2014; Papernot et al., 2016). Since then, a wide variety of deep learning tasks have been shown to also be vulnerable to adversarial attacks, ranging from various computer vision tasks to natural language processing tasks (Jia & Liang, 2017; Zhang et al., 2020; Jin et al., 2020; Alzantot et al., 2018).

Perhaps unsurprisingly, deep reinforcement learning (DRL) agents are also vulnerable to adversarial attacks, as first shown in (Huang et al., 2017) for atari games DRL agents. (Huang et al., 2017) study the effectiveness of adversarial examples on a policy network trained on Atari games under the situation where the attacker has access to the neural network of the victim policy. In (Lin et al., 2017), the authors further investigate a strategically-timing attack when attacking victim agents on Atari games at a subset of the time-steps. Meanwhile, (Kos & Song, 2017) use the fast gradient sign method (FGSM) (Goodfellow et al., 2014) to generate adversarial perturbation on the A3C agents (Mnih et al., 2016) and explore training with random noise and FGSM perturbation to improve resilience against adversarial examples. While the above research endeavors focus on actions that take discrete values, another line of research tackles a more challenging problem on DRL with continuous action spaces (Weng et al., 2019; Gleave et al., 2019). Specifically, (Weng et al., 2019) consider a two-step algorithm which determines adversarial perturbation to be closer to a targetted failure state using a learnt dynamics model, and (Gleave et al., 2019) propose a physically realistic threat model and demonstrate the existence of adversarial policies in zero-sum simulated robotics games. However, all the above works focused on the *single* DRL setting.

While most of the existing DRL attack algorithms focus on the *single* DRL agent setting, in this work we propose to study the vulnerability of *multi-agent* DRL, which has been widely applied in many safety-critical real-world applications including swarm robotics (Dudek et al., 1993), electricity distribution, and traffic control (OroojlooyJadid & Hajinezhad, 2019). In particular, we focus on the collaborative multi-agent reinforcement learning (c-MARL) setting, where a group of agents is trained to generate joint actions to maximize the team reward. We note that c-MARL is a more challenging yet interesting setting than the *single* DRL agent setting, as now one also needs to consider the interactions between agents, which makes the problem becomes more complicated.

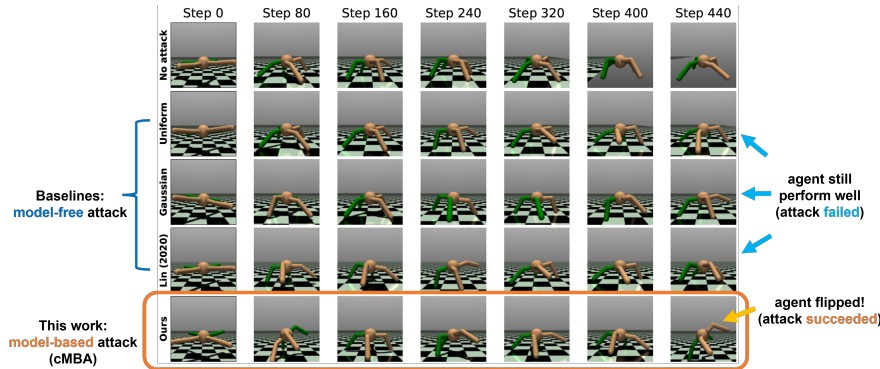

Figure 1: We illustrate the proposed model-based attack is powerful while other model-free attacks failed on attacking Agent 0 in **Ant (4x2)** environment. The episode ends after 440 time steps under our **c-MBA** (the agent flips), demonstrating the effectiveness of our algorithm.

Our contribution can be summarized as follows:

- In this work, we propose the first *model-based* adversarial attack framework on c-MARL, where we name it **c-MBA** (**M**odel-**B**ased **A**ttack on **c**-MARL). We formulate the attack into a two-step process and solve for adversarial state perturbation efficiently by existing proximal gradient methods. We show that our model-based attack is stronger and more effective than all of existing model-*free* baselines. Besides, we propose a novel adaptive victim selection strategy and show that it could further increase the attack power of **c-MBA** by decreasing the team reward even more.

- To alleviate the dependence on the knowledge of the c-MARL environment, we also propose the first data-driven approach to define the targeted failure state based on the collected data for training the dynamics model. Our numerical experiments illustrate that **c-MBA** with the data-driven failure state is comparable and even outperforms **c-MBA** with the expert-defined failure state in many cases. Therefore, our data-driven approach is a good proxy to the optimal failure state when we have little or no knowledge about the state space of the c-MARL environments.

- We show on both the multi-agent MuJoCo and multi-agent particle environments that our **c-MBA** consistently outperforms the SOTA baselines in all tested environments. We show that c-MBA can reduce the team reward up to $8-9\times$ when attacking the c-MARL agents. In addition, **c-MBA** with the proposed victim selection strategy matches or even outperforms other **c-MBA** variants in all environments with up to 80% of improvement on reducing team reward.

**Paper outline.** Section 2 discusses related works in adversarial attacks for DRL and present general background in c-MARL setting. We describe our proposed attack framework **c-MBA** in Section 3.1 for a fixed set of victim agents. In addition, we propose an alternative data-drive approach to determine the failure state for c-MBA in Section 3.2. After that, we detail an adaptive strategy to design a stronger attack by selecting the most vulnerable victim agents in Section 3.3. Section 4 presents the evaluation of our approach on several standard c-MARL benchmarks. Finally, we summarize our results and future directions in Section 5.

## 2 RELATED WORK AND BACKGROUND

**Related work.** Most of existing adversarial attacks on DRL agents are on *single* agent (Huang et al., 2017; Lin et al., 2017; Kos & Song, 2017; Weng et al., 2019) while there is only two other works (Lin et al., 2020; Hu & Zhang, 2022) that focus on the c-MARL setting. Whereas (Hu & Zhang, 2022) considers a different problem than ours where they want to find an optimally "sparse" attack by finding an attack with minimal attack steps, (Lin et al., 2020) proposes a two-step attack procedure to generate state perturbation for c-MARL setting which is the most relevant to our work.

However, there are two major differences between our work and (Lin et al., 2020): (1) their attack is only evaluated under the StarCraft Multi-Agent Challenge (SMAC) environment (Samvelyan et al., 2019) where the action spaces are discrete; (2) their approach is model-*free* as they do not involve learning the dynamics of the environment and instead propose to train an adversarial policy for a fixed agent to minimize the the total team rewards. The requirement on training an adversarial policy is impractical and expensive compared to learning the dynamics model. To the best of our knowledge, there has not been any work considering adversarial attacks on the c-MARL setting using model-based approach on continuous action spaces. In this paper, we perform adversarial attacks on agents trained using MADDPG (Lowe et al., 2017) on two multi-agent benchmarks including multi-agent MuJoCo and multi-agent particle environments. Note that in the setting of adversarial attacks, once the agents are trained, policy parameters will be frozen and we do not require any retraining of the c-MARL agents during our attack.

**Background in c-MARL.** We consider multi-agent tasks with continuous action spaces modeled as a Decentralized Partially Observable Markov Decision Process (Dec-POMDP) (Oliehoek & Amato, 2016). A Dec-POMDP has a finite set of agents $\mathcal{N} = \{1, \cdots, n\}$ associated with a set of states $\mathcal{S}$ describing global states, a set of continuous actions $\mathcal{A}_i$, and a set of individual state $\mathcal{S}_i$ for each agent $i \in \mathcal{N}$. Given the current state $s_t^i \in \mathcal{S}_i$, the action $a_t^i \in \mathcal{A}_i$ is selected by a parameterized policy $\pi^i : \mathcal{S}_i \to \mathcal{A}_i$. The next state for agent $i$ is determined by the state transition function $\mathcal{P}_i : \mathcal{S} \times \mathcal{A}_i \to \mathcal{S}$, and agent $i$ will receive a reward $r_t^i$ calculated from a reward function $\mathcal{R}_i : \mathcal{S} \times \mathcal{A}_i \to \mathbb{R}$ and observe a new state $s_{t+1}^i \in \mathcal{S}_i$. In addition, Dec-POMDP is associated with an initial state distribution $\mathcal{P}_0$ and a discount factor $\gamma$. Training a c-MARL agent is to find a joint policy that maximize the total team rewards $\sum_{i,t} r_t^i$. Note that for the ease of exposition, we do not differentiate state and observation in this work and use them interchangeably throughout the paper.

## 3 C-MBA: MODEL-BASED ATTACK FOR C-MARL

### 3.1 PROBLEM FORMULATION AND C-MBA ATTACK

Our goal is to generate adversarial perturbations imposed to the victim agents' input (state) in order to deteriorate the total team reward. The added perturbations encourages the victim agents' state to be close to a desired failure state corresponding to low reward. To avoid sampling from the environment, we use a pre-trained model that learns the dynamics of the environment to predict the next state from the perturbed state and current action, then find the suitable noise that minimizes the distance between the predicted next state and a predefined target state. For now, we assume the target state is given and we show in Section 3.2 that this target state can actually be learned directly from the data. The overall attack can be formulated as an optimization problem as follows.

Formally, we consider a multi-agent setting with $|\mathcal{N}| = n$ agents, each agent $i \in \mathcal{N}$ receives state $s_t^i$ locally and takes action $a_t^i$ following the pre-trained c-MARL policy $\pi^i(s_t^i)$. Let $s_t = (s_t^1, \cdots, s_t^n) \in \mathcal{S}$ be the joint global state at time step $t$ which is concatenated from local states $s_t^i$ for each agent $i \in \mathcal{N}$. We also denote the joint action $a_t = (a_t^1, \cdots, a_t^n)$ concatenated from each agent's action $a_t^i$. Let $\mathcal{V}_t \subseteq \mathcal{N}$ be the set of victim agents at time step $t$, i.e. the set of agents that can be attacked. Let $f : \mathcal{S} \times \mathcal{A} \to \mathcal{S}$ be a parameterized function that approximates the dynamics of the environment, where $\mathcal{A}$ is the set of concatenated actions, one from each $\mathcal{A}_i$. Let $s_{fail}$ be the targeted failure state which can lead to poor performance to the agent. We denote $\varepsilon$ as an upper bound on budget constraint w.r.t some $\ell_p$-norm $\|\cdot\|_p$. The state perturbation $\Delta s = (\Delta s^1, \cdots, \Delta s^n)$ (we suppress the dependence on $t$ of $\Delta s$ to avoid overloading the notation) to $s_t$ is the solution to the following problem:

$$
\begin{aligned}
\min_{\Delta s = (\Delta s^1, \cdots, \Delta s^n)} \quad & d(\hat{s}_{t+1}, s_{fail}) \\
\text{s.t.} \quad & \hat{s}_{t+1} = f(s_t, a_t) \\
& a_t^i = \pi^i(s_t^i + \Delta s^i), \quad \forall i \in \mathcal{N} \\
& \Delta s^i = \mathbf{0}, \quad \forall\, i \notin \mathcal{V}_t \\
& \ell_{\mathcal{S}} \le s_t + \Delta s \le u_{\mathcal{S}} \\
& \|\Delta s^i\|_p \le \varepsilon, \quad \forall i \in \mathcal{V}_t
\end{aligned}
\tag{1}
$$

where $\mathbf{0}$ is a zero vector, and the state vector follows a boxed constraint specified by $\ell_{\mathcal{S}}$ and $u_{\mathcal{S}}$.

Let us first provide some insights for the formulation (1). For each agent $i$, using the trained policy $\pi^i$, we can compute the corresponding action $a_t^i$ given its (possibly perturbed) local state $s_t^i$ or $s_t^i + \Delta s^i$. From the concatenated state-action pair $(s_t, a_t)$, we can predict the next state $\hat{s}_{t+1}$ via the learned dynamics model $f$. Then by minimizing the distance between $\hat{s}_{t+1}$ and the targeted failure state $s_{fail}$ subject to the budget constraint, we are forcing the victim agents to move closer to a damaging failure state in the next time step leading to low team reward.

Note that problem (1) can be reformulated as a constrained nonconvex problem which can be efficiently solved by first-order method to obtain a stationary point. We defer the details to Appendix A. Here, the convergence guarantee is that the perturbation found by solving the optimization problem (1) will make the next state (predicted by our dynamics model) closest to the failure state given the budget constraint. Finally, the full attack algorithm of c-MBA at timestep $t$ can be summarized in Alg. 1.

---

**Algorithm 1** c-MBA algorithm at timestep $t$

---

1: **Initialization:**
2:     Given $s_t$, $s_{fail}$, $\pi$, $f$, $\mathcal{V}_t$; initialize $\Delta s = \varepsilon * sign(x)$ for $x \sim N(0, 1)$, attack budget $\varepsilon$, $p$; choose learning rate $\eta > 0$
3: **For** $k = 0, \cdots, K - 1$ **do**
4:     Compute $a_t = (a_t^1, \cdots, a_t^n)$ where $a_t^i = \pi^i(s_t^i + \Delta s^i)$ if $i \in \mathcal{V}_t$ and $a_t^i = \pi^i(s_t^i)$ otherwise.
5:     Compute $\hat{s}_{t+1} = f(s_t, a_t)$.
6:     Update $\Delta s$ as $\Delta s_{k+1} = \text{proj}_{\mathcal{C}_{p,\varepsilon,t}} [\Delta s_k - \eta \nabla_{\Delta s} d(\hat{s}_{t+1}, s_{fail})]$.
7: **End For**

---

**Learning dynamics model.** One of the key enabler to solve (1) is the availability of the learned dynamics model $f$. If the dynamics is known, we can solve (1) easily with proximal gradient methods. However, in practice we often do not have the full knowledge of the environment and thus if we would like to solve (1), we can learn the dynamics model via some function approximator such as neural networks. The parameter $w$ for $f$ is the solution of the following optimization problem

$$\min_{\phi} \sum_{t \in \mathcal{D}} \|f(s_t, a_t; \phi) - s_{t+1}\|^2 \tag{2}$$

where $\mathcal{D}$ is a collection of state-action transitions $\{(s_t, a_t, s_{t+1})\}_{t \in \mathcal{D}}$ and $s_{t+1}$ is the actual state that the environment transitions to after taking action $a_t$ determined by a given policy. In particular, we separately collect transitions using the pre-trained policy $\pi_{tr}$ and a random policy $\pi_{rd}$ to obtain $\mathcal{D}_{train}$ and $\mathcal{D}_{random}$. The motivation of using the random policy to sample is to avoid overfitting the dynamics model to the trained policy. Then the dataset $\mathcal{D}$ is built as $\mathcal{D} = \mathcal{D}_{train} \cup \mathcal{D}_{random}$. Since (2) is a standard supervised learning problem, the dynamics model $f$ can be solved by existing gradient-based methods. We describe the full process of training the dynamics model in Alg. 4 where the `GradientBasedUpdate` step could be any gradient-descent-type update. In our experiments, we notice that the quality of the dynamics model does not significantly affect the result as seen in Figure 11. We find that even when we use a less accurate dynamics model, our proposed c-MBA attack is still effective. Our attack using the dynamics model trained with only 0.2 M samples for only 1 epoch is comparable with ones using more accurate dynamics model (1M for 1 and 100 epochs) in the **Ant (4x2)** environment under both $\ell_\infty$ and $\ell_1$ budget constraint. We describe the whole process of training the dynamics model in Algorithm 4 in Appendix B.

**Discussion: difference between the baseline (Lin et al., 2020).** We note that the most closely related to our work is (Lin et al., 2020), where they also propose an attack algorithm to destroy c-MARL. However, there are two major differences between their approach and ours: (1) Their method is a *model-free* approach based on training extra adversarial policies, which could be impractical as it requires a lot of samples and it may be difficult to collect the required "bad" trajectories to minimize the team reward (this requires the full control of all the agents in the c-MARL setting which may not be available in practice). On the other hand, our c-MBA is a *model-based* approach, where we only need to have a rough surrogate of the environment. This is an more practical scenario, and even very crude dynamics model could make c-MBA effective (see **Experiment (III)** in Section 4). (2) They did not leverage the unique setting in MARL to select most vulnerable agent to attack, while in the next section, we show that with the victim agent selection strategy, we could make c-MBA an even stronger attack algorithm (also see **Experiment (IV)** for more details).

**Remark 3.1** *We note that our approach is applicable for the **black-box attacks** where an adversary does not have access to the policy parameter. The reason is because Alg. 1 and 3 only need to have a mechanism to access the gradient of the output action with respect to the state input, which could be estimated by finite difference methods similar to the black-box attacks in the image classification setting. As the baseline method in (Lin et al., 2020) focuses on white-box attack, we also implement our approach under the white-box setting for fair comparison.*

## 3.2 Learning failure states from data

In order to solve (1), we need to specify the failure state $s_{fail}$ which often requires prior knowledge of the state definition of the c-MARL environment. To make our method more flexible, we propose the first data-driven approach to learn the failure state. As our c-MBA attack involves training a dynamics model of the environment by collecting the transition data, we can learn the failure state from directly the pre-collected dataset without extra overhead. Based on the observation that the a failure state should be a resulting state where the reward corresponding to that transition is low, we sort the collected transition $(s_t, a_t, \hat{s}_t, r_t)$ by ascending order of the reward $r_t$ and choose the failure state to be $\hat{s}_s^i$ that corresponds to the lowest $r_t$ in the dataset. This process is described in Algorithm 2:

---
**Algorithm 2** Learning failure state from collected data
---
1: **Input::** a dataset $\mathcal{D} = \{(s_t, a_t, s_{t+1})\}_{t \in \mathcal{D}}$ as a collection of transitions.
2:    Sort the transition by ascending order of reward $r_t$
3:    Determine $(s_{(1)}, a_{(1)}, \hat{s}_{(1)}, r_{(1)})$ as the transition corresponding to the minimum reward.
4:    Set $s_{fail} = \hat{s}_{(1)}$.

---

Using this data-driven strategy, we show that performing c-MBA attack with the learned failure state either matches or performs better than the expert-defined failure state for multi-agent MuJoCo environments. In addition, the data-driven approach demonstrates its advantage in the multi-agent particle environment where we do not have expert knowledge of the state space. Overall, c-MBA using the learned failure state shows its superior performance over other model-free baselines in most cases in the experiments.

## 3.3 Crafting stronger attack with c-MBA – victim agent selection strategy

In this subsection, we propose a new strategy to select most vulnerable victim agents with the goal to further increase the power of our cMBA. We note that this scenario is unique in the setting of *multi-agent* DRL setting, as in the *single* DRL agent setting we can only attack the same agent all the time. To the best of our knowledge, our work is the first to consider the victim selection strategy as (Lin et al., 2020) only use one fixed agent to perform the attack. As a result, we can develop a stronger attack by selecting appropriate set of "vulnerable" agents. This strategy can be effective in the "sparse attack" setting when only a few agents in the team are attacked (Hu & Zhang, 2022). To start with, we first formulate a mixed-integer program to perform the attack on a set of victim agents as below:

$$
\begin{aligned}
\min_{\Delta s, w} \quad & d(\hat{s}_{t+1}, s_{fail}) \\
\text{s.t.} \quad & \hat{s}_{t+1} = f(s_t, a_t) \\
& a_t^i = \pi_i(s_t^i + w_i \cdot \Delta s^i), \quad \forall i \in \mathcal{N} \\
& \ell_{\mathcal{S}} \le s_t^i + \Delta s^i \le u_{\mathcal{S}}, \quad \forall i \in \mathcal{N} \\
& \left\| \Delta s^i \right\|_p \le \varepsilon, \quad \forall i \in \mathcal{N} \\
& w_i \in \{0, 1\}, \quad \forall i \in \mathcal{N} \\
& \sum_i w_i = n_v
\end{aligned}
\tag{3}
$$

where we introduce a new set of binary variables $\{w_i\}$ to select the suitable agent to attack.

Due to the existence of the new binary variables, problem (3) is much harder to solve. Therefore, we instead propose to solve a proxy of (3) as follows

$$
\begin{aligned}
\min_{\Delta s, \theta} \quad & d(\hat{s}_{t+1}, s_{fail}) \\
\text{s.t.} \quad & \hat{s}_{t+1} = f(s_t, a_t) \\
& a_t^i = \pi^i(s_t^i + W_i(s_t; \theta) \cdot \Delta s^i), \quad \forall i \in \mathcal{N} \\
& \ell_{\mathcal{S}} \leq s_t^i + \Delta s^i \leq u_{\mathcal{S}}, \quad \forall i \in \mathcal{N} \\
& \left\| \Delta s^i \right\|_p \leq \varepsilon, \quad \forall i \in \mathcal{N} \\
& 0 \leq W_i(s_t; \theta) \leq 1, \quad \forall i \in \mathcal{N}
\end{aligned}
\tag{4}
$$

where $W(s; \theta) : \mathcal{S} \to \mathbb{R}^n$ is a function parametrized by $\theta$ that takes current state $s$ as input and returns the weight to distribute the noise to each agent. Suppose we represent $W(s; \theta)$ by a neural network, we can rewrite the formulation (4) as (5) because the last constraint in (4) can be enforced by using a softmax activation in the neural network $W(s; \theta)$

$$
\begin{aligned}
\min_{\Delta s, \theta} \quad & d(f(s_t, \pi(\{s_t^i + W_i(s_t; \theta) \cdot \Delta s^i\}_i)), s_{fail}) \\
\text{s.t.} \quad & \Delta s \in \mathcal{C}_{p, \varepsilon, t}
\end{aligned}
\tag{5}
$$

where the notation $\pi(\{s_t^i + W_i(s_t; \theta) \cdot \Delta s^i\}_i)$ denotes the concatenation of action vectors from each agent to form a global action vector $[\pi^1(s_t^1 + W_1(s_t; \theta) \cdot \Delta s^1), \ldots, \pi^n(s_t^n + W_n(s_t; \theta) \cdot \Delta s^n)]$. As a result, (5) can be efficiently solved by using PGD. We present the pseudo-code of the attack in Alg. 3. After $K$ steps of PGD update, we define the $i_{(n-j+1)}$ as index of the $j$-th largest value within $W(s_t; \theta_K) \in \mathbb{R}^n$, i.e. $W_{i_{(n)}}(s_t; \theta_K) \geq W_{i_{(n-1)}}(s_t; \theta_K) \geq \cdots \geq W_{i_{(1)}}(s_t; \theta_K)$. Let $\mathcal{I}_j$ be the index set of top-$j$ largest outputs of the $W(s_t; \theta_K)$ network. The final perturbation returned by our victim agent selection strategy will be $\widehat{\Delta s} = ((\widehat{\Delta s})^1, \cdots, (\widehat{\Delta s})^n)$ where $(\widehat{\Delta s})^i = \mathbf{0}$ if $i \notin \mathcal{I}_{n_v}$ and $(\widehat{\Delta s})^i = (\Delta s_K)^i$ if $i \in \mathcal{I}_{n_v}$.

---

**Algorithm 3** cMBA with victim agent selection at time-step $t$

---

1: **Initialization:** Given $s_t, s_{fail}, \pi, f, n_v$; initialize $\Delta s_0$; choose learning rate $\eta, \lambda > 0$.
2: **For** $k = 0, \cdots, K - 1$ **do**
3:     Compute $a_t = \pi(\{s_t^i + W_i(s_t; \theta) \cdot \Delta s^i\}_i)$.
4:     Compute $\hat{s}_{t+1} = f(s_t, a_t)$.
5:     Update $\Delta s$ as $\Delta s_{k+1} = \text{proj}_{\mathcal{C}_{p, \varepsilon, t}} [\Delta s_k - \eta \nabla_{\Delta s} d(\hat{s}_{t+1}, s_{fail})]$.
6:     Update $\theta$ as $\theta_{k+1} = \theta_k - \lambda \nabla_\theta d(\hat{s}_{t+1}, s_{fail})$.
7: **End For**
8: Compute $\mathcal{I}_{n_v} = \{i_{(n)}, \cdots, i_{(n-n_v)}\}$ such that $W_{i_{(n)}}(s_t, \theta_K) \geq \cdots \geq W_{i_{(1)}}(s_t, \theta_K)$.
9: Return $\widehat{\Delta s} = ((\widehat{\Delta s})^1, \cdots, (\widehat{\Delta s})^n)$ where $(\widehat{\Delta s})^i = (\Delta s_K)^i$ if $i \in \mathcal{I}_{n_v}$ and $(\widehat{\Delta s})^i = 0$ otherwise.

---

**Remark 3.2** *For this attack, we assume each agent $i$ has access to the other agent's state to form the joint state $s_t$. If the set of victim agents is pre-selected, we do not need this assumption and the adversarial attack can be performed at each agent independently of others.*

**Remark 3.3** *We can also use Alg. 3 in conjunction with Alg. 1 to perform the attack using Alg. 1 on the set of victim agents returned by Alg. 3. Our results for Experiment (IV) in Section 4 indicate that using Alg. 3 and Alg. 1 together produces an even stronger cMBA attack: e.g. it can further decrease the team reward up to 267% more than using only Alg. 3 in **Ant (4x2)** environment.*

We highlight that combining cMBA with victim agent selection strategy can potentially make cMBA much stronger as shown in our comparison with other selection strategy such as random selection or greedy selection in Section 4. Our numerical results show that selecting the appropriate set of victim agents at each time-step constantly outperform cMBA attack with random/greedy agent selection. For example, with victim agent selection strategy, cMBA can further lower the team reward up to 267%, 486%, 30% *more* than when using cMBA on randomly selected, greedily selected, or fixed agent in **Ant (4x2)** environment.

## 4    EXPERIMENTS

We perform the attack on both multi-agent MuJoCo (MA-MuJoCo) environments (de Witt, 2020) including **Ant(4x2)**, **HalfCheetah(2x3)**, **HalfCheetah(6x1)**, and **Walker2d(2x3)**. The pair **name(config)** indicates the name of MuJoCo environment along with the agent partition, where a configuration of 2x3 means there are in total 2 agents and each agent has 3 actions. We provide more details on the construction of these agents in Appendix C. We also demonstrate the effectiveness of the learned failure state approach using the multi-agent particle environment, denoted as **MPE(3x5)** (Lowe et al., 2017; Mordatch & Abbeel, 2017) where we do not have expert knowledge of the failure state. We consider the following **model-free** baselines:

1. **Uniform**: the perturbation follows the Uniform distribution $U(-\varepsilon, \varepsilon)$.
2. **Gaussian**: the perturbation follows the Normal distribution $\mathcal{N}(0, \varepsilon)$.
3. **Lin et al. (2020) + iFGSM**: Since there is no other work performing adversarial attack for continuous action space in c-MARL, we adapt the approach in (Lin et al., 2020) to form another baseline. In particular, we train an adversarial policy for one agent to minimize the total team reward while the remaining agents use the trained MARL policy. This adversarial policy is trained for 1 million timesteps. We then use this trained policy to generate a "target" action and use iterative FGSM method (Kurakin et al., 2016; Goodfellow et al., 2014) to generate the adversarial observation perturbation for the agents' input. Note that the adversarial policy is trained on the same agent that is being attacked.

In our experiments, we consider two variants of c-MBA as follows.

1. **c-MBA-F**: we perform c-MBA using an expert-defined failure state where we have the knowledge of the state definition of the c-MARL environment. This variant is used on the MA-MuJoCo benchmark as there are available documentation (AI) on its state definition.

2. **c-MBA-D**: we perform c-MBA attack using failure state learned from the collected data as described in Section 3.2.

Due to space constraint, we only present part of our results here and we defer the full results in Appendix D. We evaluate c-MBA comprehensively with the following 5 experiments:

- **Experiment (I) – model-free baselines vs model-based attack c-MBA using $\ell_\infty$-constrained perturbation**: we compare c-MBA-F and c-MBA-D with other baselines when attacking individual agent under $\ell_\infty$ constraint.
- **Experiment (II) – effectiveness of learned adaptive victim selection**: we evaluate the performance of Alg. 3 with other heuristicly selected variants of c-MBA. Our results show that in **HalfCheetah(6x1)** environment, under $\varepsilon = 0.05$ and $\ell_\infty$ budget constraint, the amount of team reward reduction of our learned agents selection scheme can be up to 80% *more* than the cases when using heuristic strategy to select victim agent.
- **Experiment (III) – attacking multiple agents using model-free baselines vs model-based attack c-MBA with $\ell_\infty$ perturbation**: we report results on attacking multiple agents simultaneously. This setting is not previously considered in (Lin et al., 2020).
- **Experiment (IV) – model-free baselines vs model-based attack c-MBA in MPE(3x5) environment**: we compare c-MBA-D with other baselines under $\ell_\infty$ and $\ell_1$ constraint.
- **Experiment (V) – model-free baselines vs model-based attack c-MBA on $\ell_1$ perturbation**: we compare c-MBA-F and c-MBA-D with other baselines when attacking individual agent under $\ell_1$ constraint. The results of this experiment are presented in Appendix D.
- **Experiment (VI) – adversarial attacks using dynamics model with various accuracy**: we illustrate the performance of c-MBA when using dynamics model with various accuracy. This experiment is deferred to Appendix D.

**Experiment (I) – model-free baselines vs model-based attack c-MBA on $\ell_\infty$ perturbation.** In this experiment, we run the 3 baseline attacks along with two variants of our model-based attack on the four MA-MuJoCo environments with one victim agent ($n_v = 1$) in coherence with (Lin et al., 2020).Fig. 2 illustrate the performance when we perform these attacks on each agent with different attack budget using $\ell_\infty$-norm. For a fixed agent, our model-based attack outperforms all

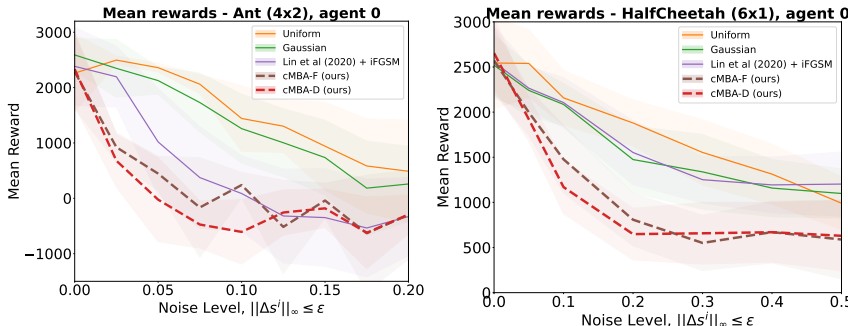

Figure 2: c-MBA vs baselines when attacking one agent in 2 MA-MuJoCo environments - **Exp. (I)**.

the other baselines. In particular, our model-based attack yields much lower rewards under relatively low budget constraints (when $\varepsilon \in [0.05, 0.2]$) compared to other baselines. For instance, under budget level $\varepsilon = 0.05$, the amount of team reward reduction of our c-MBA attack is (on average) 66%, 503%, and 806% *more* than the three model-free (Lin et al. (2020) + iFGSM, Gaussian, and Uniform, respectively) in **Ant(4x2)** environment. We also observe that c-MBA-D either matches or performs better than c-MBA-F with the expert-defined failure state.

**Experiment (II) – effectiveness of learned adaptive victim selection.** To evaluate our proposed victim-agent selection strategy, we consider the following variants of our model-based attack:

1. **c-MBA(fixed agents)**: attack a fixed set of victim agents with Alg. 1. We use **c-MBA (best fixed agents)** to denote the best result among fixed agents.
2. **c-MBA(random agents)**: uniformly randomly select victim agents to attack with Alg. 1.
3. **c-MBA(greedy agents selection)**: for each time step, sweep all possible subsets of agents with size $n_v$ and perform Alg. 1. Select the final victim agents corresponding to the objective value (distance between predicted observation and target failure observation).
4. **c-MBA(learned agents selection)**: use Alg. 3 to attack the most vulnerable agents.
5. **c-MBA(learned agents selection + Alg. 1)**: use Alg. 3 to select the most vulnerable agents to attack then perform the attack with the selected agents with Alg. 1.

We conducted experiments on all 4 MA-MuJoCo environments and part of the results are plotted in Fig. 3. It shows that **c-MBA(learned agents selection)** and **c-MBA(learned agents selection + Alg. 1)** perform better than other variants in both environments. It is interesting to observe that **c-MBA(learned agents selection + Alg. 1)** is either comparable or better than **c-MBA(learned agents selection)** which shows that running Alg. 1 using agents selected by Alg. 3 can be beneficial. We show that the victim selection technique is really important as randomly choosing the agents to attack (i.e. **c-MBA(random agents)**) or even choosing agents greedily (i.e. **c-MBA(greedy agents selection)**) cannot form an effective attack and are in fact, even worse than the attacking on fixed agents (i.e **c-MBA (best fixed agents)**) in most cases. For example, our results show that in **HalfCheetah(6x1)** environment, under $\varepsilon = 0.05$ and $\ell_\infty$ budget constraint, the amount of team reward reduction of our learned agents selection scheme is 33%, 80%, 35% *more* than the cases when attacking fixed agents, random agents, or greedily selected agents, respectively.

**Experiment (III) – attacking two agents using model-free baselines vs model-based attack c-MBA using $\ell_\infty$ constrained:** We conduct experiments using model-free and model-based approaches to simultaneously attack two agents in Ant(4x2) environment. Fig. 4 illustrate the performance of various attacks. We observe that our c-MBA attack outperforms other baselines. For instance, at $\varepsilon = 0.025$, the team reward reduction of c-MBA is 68%, 522%, 713% *more* than the three model-free baselines. In addition, our c-MBA-D even outperforms c-MBA-F to achieve lower reward at all budget levels.

**Experiment (IV) – model-free baselines vs model-based attack c-MBA in MPE(3x5) environment**: We perform the same procedure as in Experiment (III) to attack different agents in the **MPE(3x5)** environments where we attack two or three agents at the same time. Since we do not have expert knowledge about the failure state in this environment, we compare **c-MBA-D** with other model-free baselines. The results are depicted in Figure 5 where **c-MBA-D** perform slightly better

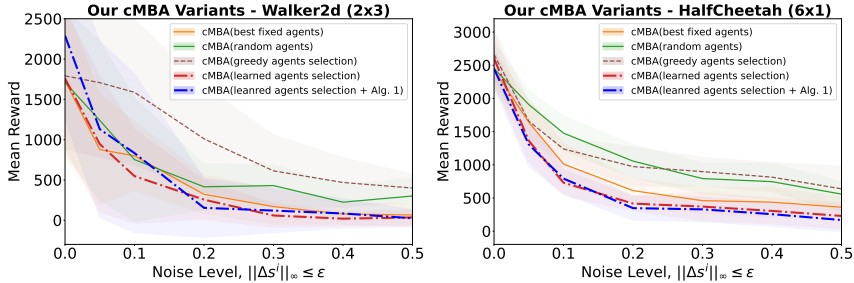

Figure 3: Performance of c-MBA with different victim agent selection strategies - **Exp. (II)**.

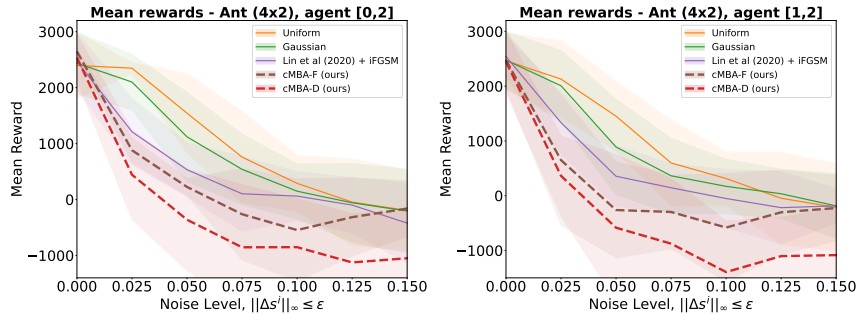

Figure 4: c-MBA vs baselines in Ant(4x2) when attacking two agents - **Exp. (III)**.

than other random noise baselines when attacking 2 agents and significantly outperforms other baselines when attacking three agents simultaneously. We note that (Lin et al., 2020) does not perform well in this experiment as we observe during training the adversarial policy that it could not lower the team reward effectively.

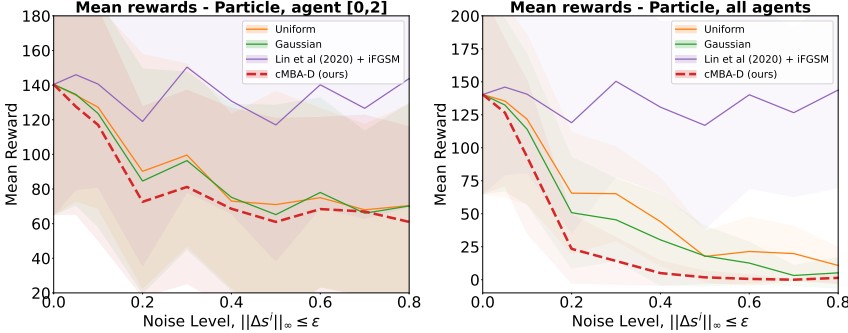

Figure 5: c-MBA vs baselines in **MPE(3x5)** when attacking two agents (3 leftmost) and three agents (rightmost) simultaneously using $\ell_\infty$ constraint - **Exp. (IV)**.

## 5 CONCLUSIONS

In this paper, we propose a new attack algorithm named **c-MBA** for evaluating the robustness of c-MARL environment. Our c-MBA algorithm is the first model-based attack to craft adversarial observation perturbations and we have shown that c-MBA outperforms existing model-free baselines attack by a large margin under both multi-agent MuJoCo and multi-agent particle benchmarks. Unique to multi-agent setting, we also propose a new victim-selection strategy to select the most vulnerable victim agents given the attack budget, which has not been studied in prior work. We show that with the victim-agent selection strategy, c-MBA attack becomes stronger and more effective. We also propose the first data-driven approach to determine the failure state based on the pre-collected data without extra overhead making our attack more flexible.

## REPRODUCIBILITY STATEMENT

We present additional numerical results in Appendix C. We also provide an example of how the state or adversarial state perturbation evolves in one of MA-MuJoCo environment. To promote reproducibility, we have uploaded the source code with full instructions to reproduce the results in this paper. We also add an example notebook to visualize the results. If our paper gets accepted, we will make our code publicly.

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

# Appendix

## A    DETAILS ON HOW TO SOLVE (1) EFFICIENTLY

Problem (1) can be efficiently solved by proximal-gradient-based methods (Nesterov, 2003). Firstly, by substituting $a_t$ and $\hat{s}_{t+1}$ with their definitions, (1) is equivalent to

$$
\begin{aligned}
\min_{x} \quad & d(f(s_t, \pi(s_t + x)), s_{fail}) \\
\text{s.t.} \quad & x \in \mathcal{C}_{p,\varepsilon,t}
\end{aligned}
\tag{6}
$$

where $\mathcal{C}_{p,\varepsilon,t} := \{x = (x^1, \cdots, x^n) : \ell_{\mathcal{S}} - s_t \leq x \leq u_{\mathcal{S}} - s_t, \left\|x^i\right\|_p \leq \varepsilon \text{ for } i \in \mathcal{V}_t, \text{ and } x^i = \mathbf{0} \text{ for } i \notin \mathcal{V}_t\}$. When $f$ and $\pi$ are represented by neural network and if we choose the distance function as $d(a,b) = \|a - b\|^2$, (6) is a constrained nonconvex problem, then We can use the projected gradient descent (PGD) algorithm (Nesterov, 2003) to solve (6). The PGD iteration to update $x_k$ at iteration $k$ starting from $x_0$ can be described as

$$
x_{k+1} = \text{proj}_{\mathcal{C}_{p,\varepsilon,t}} \left[ x_k - \eta \nabla_x d(f(s_t, \pi^i(s_t + x)), s_{fail}) \right]
$$

where $\text{proj}_{\mathcal{C}_{p,\varepsilon,t}}(\cdot)$ is the projection to the convex set $\mathcal{C}_{p,\varepsilon,t}$ and $\eta$ is the learning rate. The projection is simple to calculate since $\mathcal{C}_{p,\varepsilon,t}$ is the intersection of a $\ell_p$-norm ball and boxed constraint.

## B    DETAILS ON TRAINING DYNAMICS MODEL

---

**Algorithm 4** Training dynamics model

---

1: **Initialization:** Given pre-trained policy $\pi_{tr}$ and a random policy $\pi_{rd}$; initialize dynamics model parameter $\phi_0$.
2: Form $\mathcal{D} = \mathcal{D}_{train} \cup \mathcal{D}_{random}$ by collecting a set of transitions $\mathcal{D}_{train}$ and $\mathcal{D}_{random}$ using policy $\pi_{tr}$ and $\pi_{rd}$, respectively.
3: **For** $k = 0, 1, \cdots$ **do**

$$
\phi_{k+1} = \texttt{GradientBasedUpdate}(\mathcal{D}, \phi_k)
$$

4: **End For**

---

For each environment, we collect 1 million transitions using the trained MARL policy $\pi_{tr}$ and a random policy $\pi_{rd}$. We partition the collected data into train and test set with a ratio 90-10. The dynamics model is represented by a fully-connected neural network. The network contains 4 hidden layers with 1000 neurons at each layer and the activation function for each hidden layer is ReLU. We train the network for 100 epochs using AdamW (Loshchilov & Hutter, 2017) with early stopping where the learning rate is tuned in the set $\{0.001, 0.0005, 0.0001, 0.00005, 0.00001\}$ to obtain the model with the best test mean squared error. Please refer to our submitted code for further details.

## C    MORE DETAILS ON EXPERIMENT SETUP IN SECTION 4

**Experiment setup.** We use MADDPG (Lowe et al., 2017) to train MARL agents for the four MA-MuJoCo environments as well as the multi-agent particle environment. Using the trained agents, we collect datasets containing one million transitions to train the dynamics model for each environment. The dynamics model is a fully connected neural network with three hidden layers of 1000 neurons. We also use a fully-connected neural network for $W(s_t, \theta)$ in (4) with two hidden layers of 200

neurons. We use AdamW (Loshchilov & Hutter, 2017) as the optimizer and select the best learning rate from $\{1, 5\} \times \{10^{-5}, 10^{-4}, 10^{-3}\}$ (the best learning rate is the one achieving lowest prediction error on a test set of $80,000$ samples). For our model-based attack, we run PGD algorithm for $K = 30$ steps to solve (6) and (3). We perform each attack over 16 episodes then average the rewards. We also illustrate the standard deviation of rewards using the shaded area in the plots.

**Agent partitioning for MA-MuJoCo environments.** Each of the original MuJoCo agent in the single-agent setting contains multiple joints and the way these joints are partitioned will lead to different multi-agent configurations. These configurations are described as follows:

- **Walker (2x3)** environment: this environment has 6 joints, 3 for each leg and the whole agent is divided into 2 group of joints $\{1, 2, 3\}$ and $\{4, 5, 6\}$ representing two legs (Peng et al., 2020, Fig. 4F).
- **HalfCheetah(2x3)** environment: there are two agents, each represents a front or rear leg with joints $\{1, 2, 3\}$ and $\{4, 5, 6\}$ (Peng et al., 2020, Fig. 4C).
- **HalfCheetah(6x1)** environment: each agent represents each of the total 6 joints (Peng et al., 2020, Fig. 4D).
- **Ant(4x2)** environment: each agent controls one leg with two joints out of 4 legs (Peng et al., 2020, Fig. 4J).

**Specifying target observation for each environment.** To perform our model based attack, we need to specify a target observation that potentially worsens the total reward. Currently, we do not have a general procedure to specify this target observation. We specify the target observations based on prior knowledge about the environments as follows. In multi-agent MuJoCo environments, each agent has access to its own observation of the agent consisting the position-related and velocity-related information. The position-related information includes part of $x, y, z$ coordinates and the quarternion that represents the orientation of the agent. The velocity-related information contains global linear velocities and angular velocities for each joint in a MuJoCo agent. We refer the reader to (Todorov et al., 2012) for more information about each MuJoCo environment. Now we describe the design of this target observation for each environment as follows:

- **Walker(2x3)** environment: Since the episode ends whenever the agent falls, i.e. the $z$ coordinate falls below certain threshold. In this environment, the target observation has a value of 0 for the index that corresponds to the $z$ coordinate of the MuJoCo agent (index 0).
- **HalfCheetah(2x3)** and **HalfCheetah(6x1)** environments: As the goal is to make agent moves as fast as possible, we set the value at index corresponding to the linear velocity to 0 (index 8).
- **Ant(4x2)** environment: As the agent can move freely in a 2D-plan, we set the index corresponding to the $x, y$ linear velocities to 0 (indices 13 and 14).

## D  ADDITIONAL EXPERIMENTS

In this section, we present experimental results in addition to ones presented in Section 4.

**Full results for Experiment (I) – model-free baselines vs model-based attack c-MBA on $\ell_\infty$ perturbation** . We first show the full results on running c-MBA and three other baselines in 4 MA-MuJoCo environments under $\ell_\infty$-norm budget constraint in Fig. 6. c-MBA performs the best in all cases and significantly outperforms other baselines in majority of the cases.

To better visualize the performance difference, Fig. 1 illustrates the environment with and without attacks captured at different time-steps. From Fig. 1, our model-based attack is able to make the MuJoCo agent flip, which terminates the episode at the 409-th timestep. The results with episode length and total rewards for each variant are: No attack$(1000, 2645.64)$, Uniform$(1000, 1663.5)$, Gaussian$(891, 1548.49)$, Lin et al. (2020) + iFGSM$(1000, 902.53)$, and **Ours$(738, 366.55)$**.

**Full results of Experiment (II) – effectiveness of learned adaptive victim selection.** We conducted experiments on all 4 MA-MuJoCo environments and the results are plotted in Fig. 7. It

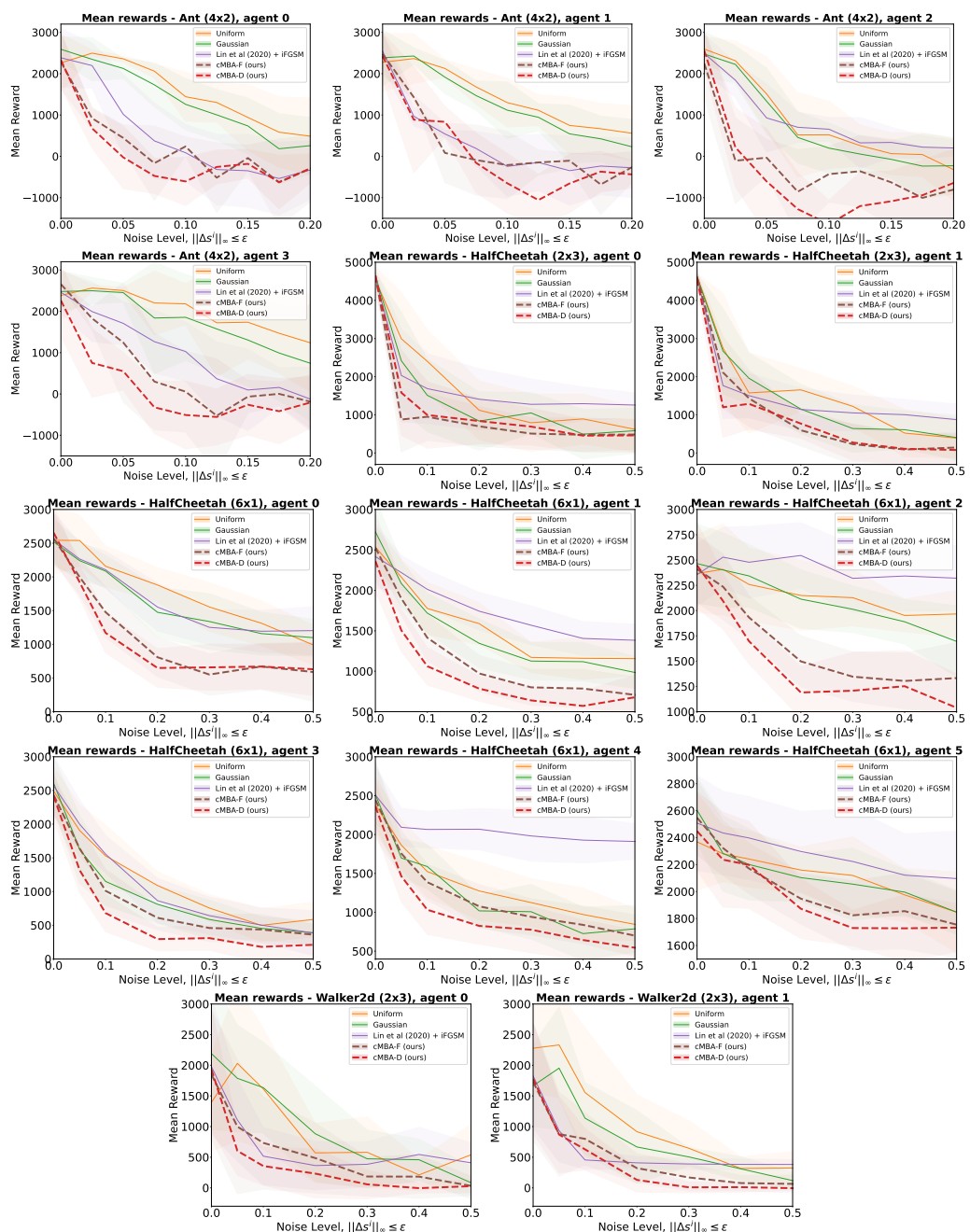

Figure 6: c-MBA vs baselines when attacking one agent in 4 MA-MuJoCo environments - **Exp. (I)**.

shows that **c-MBA(learned agents selection)** and **c-MBA(learned agents selection + Alg. 1)** appear to be better than the random or greedy strategy and is comparable with the randomly selected one in HalfCheetah(2x3) environment. It is interesting to observe that **c-MBA(learned agents selection + Alg. 1)** is either comparable or better than **c-MBA(learned agents selection)** which shows that running Alg. 1 using agents selected by Alg. 3 can be beneficial. We show that the victim selection technique is really important as randomly choosing the agents to attack (i.e. **c-MBA(random agents)**) or even choosing agents greedily (i.e. **c-MBA(random agents)**) cannot form an effective attack and are in fact, even worse than the attacking on fixed agents (i.e **c-MBA (best fixed agents)**) in most cases. For example, our results show that in **HalfCheetah(6x1)** environment, under $\varepsilon = 0.05$ and $\ell_\infty$ budget constraint, the amount of team reward reduction of our learned agents

selection scheme is 33%, 80%, 35% *more* than the cases when attacking fixed agents, random agents, or greedily selected agents, respectively.

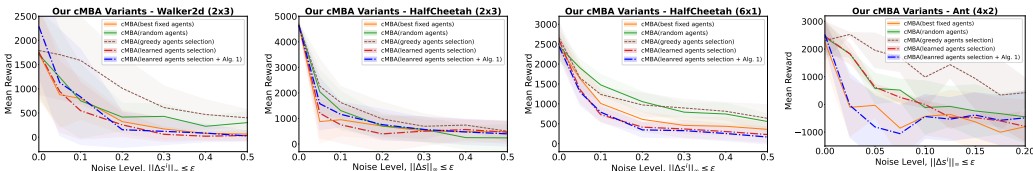

Figure 7: Performance of c-MBA with different victim agent selection strategies - **Exp. (II)**.

**Full results of Experiment (III) – attacking two agents using model-free baselines vs model-based attack c-MBA using $\ell_\infty$ constrained:** We conduct experiments using model-free and model-based approaches to simultaneously attack two agents in Ant(4x2) environment. Fig. 8 illustrate the performance of various attacks. We observe that our c-MBA attack outperforms other baselines in 5 out of 6 settings, and especially it's able to achieve low reward (close to or below 0) at lower attack budget levels. For example, at $\varepsilon = 0.025$, the team reward reduction of c-MBA is 68%, 522%, 713% *more* than the three model-free baselines.

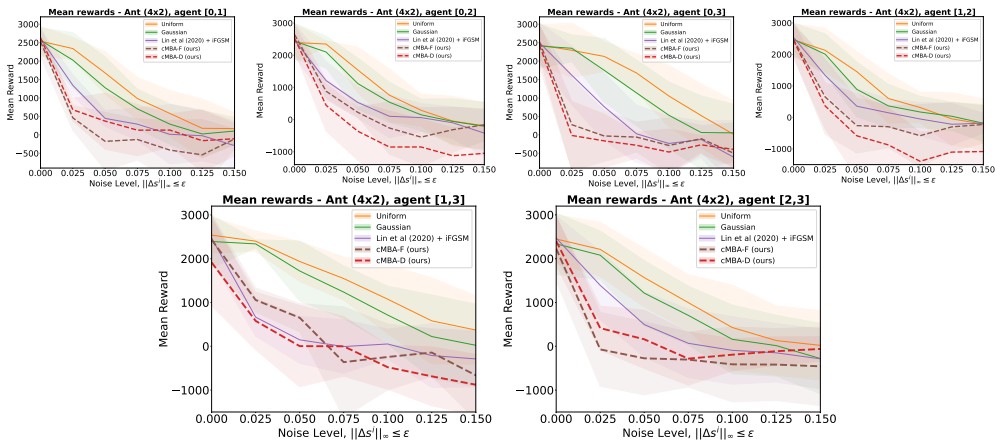

Figure 8: c-MBA vs baselines in Ant(4x2) when attacking two agents - **Exp. (III)**.

**Full resuts of Experiment (IV) – Model-free baselines vs model-based attack c-MBA in MPE(3x5) environment**: We follow the same procedure as in Experiment (III) to attack different agents in the **MPE(3x5)** environments where we attack two or three agents at the same time. Since we do not have expert knowledge about the failure state in this environment, we compare **c-MBA-D** with other model-free baselines. The results are depicted in Figure 9 where **c-MBA-D** perform slightly better than other random noise baselines when attacking 2 agents and significantly outperforms other baselines when attacking three agents simultaneously. We note that (Lin et al., 2020) does not perform well in this experiment as we observe during training the adversarial policy that it could not lower the team reward effectively.

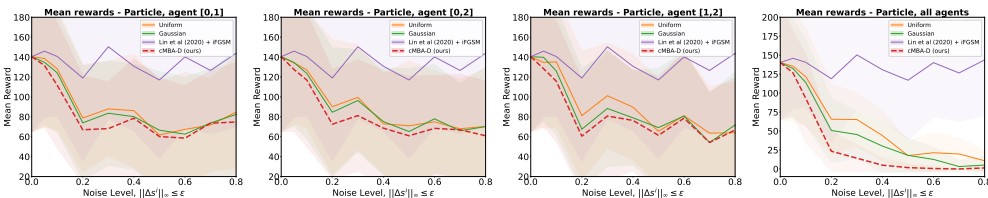

Figure 9: c-MBA vs baselines in **MPE(3x5)** when attacking two agents (3 leftmost) and three agents (rightmost) simultaneously using $\ell_\infty$ constraint - **Exp. (IV)**.

**Experiment (V): model-free baselines vs model-based attack c-MBA on $\ell_1$ perturbation.** In addition to the $\ell_\infty$-norm budget constraint, we also evaluate adversarial attacks using the $\ell_1$-norm

constraint. Note that using $\ell_1$-norm for budget constraint is more challenging as the attack needs to distribute the perturbation across all observations while in the $\ell_\infty$-norm the computation of perturbation for individual observation is independent. Fig. 10 illustrate the effect of different attacks on **HalfCheetah(6x1)** and **Walker2d(2x3)** environments, respectively. Our c-MBA is able to outperform other approaches in almost all settings. In **HalfCheetah(6x1)**, using $\varepsilon = 1.0$ under $\ell_1$ budget constraint, the amount of total team reward reduced by c-MBA variants is up to 156%, 37%, and 42% *more* than Lin et al. (2020) + iFGSM, Gaussian, and Uniform baselines, respectively.

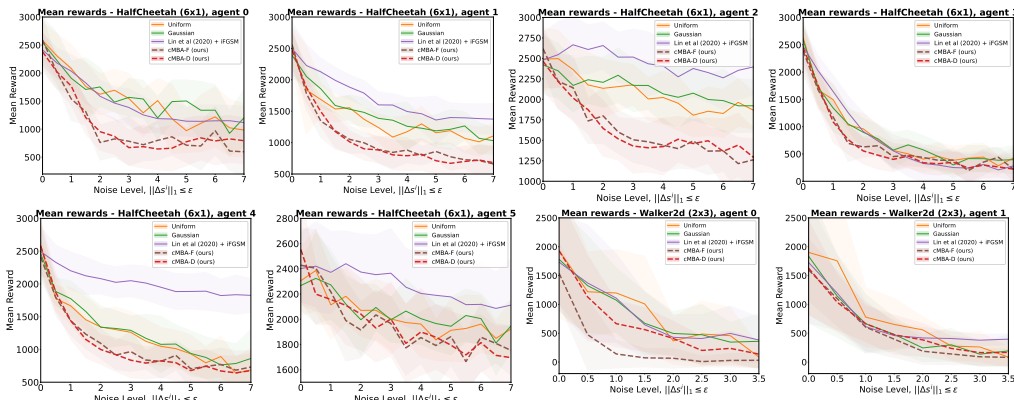

Figure 10: c-MBA vs baselines in **HalfCheetah(6x1)** and **Walker2d(2x3)** - **Exp. (V)**.

**Experiment (VI) – adversarial attacks using dynamics model with various accuracy.** We compare our attack when using trained dynamics model with different mean-squared error (MSE). We use **Ant(4x2)** environment and consider the following three dynamics models:

- **model(1M, 100 epochs)** is trained using 1 million samples for 100 epochs and select the model with the best performance.
- **model(1M, 1 epoch)** is trained using 1 million samples for only 1 epoch.
- **model(0.2M, 1 epoch)** is trained using less number of samples to only 200k and we train it for only 1 epoch.

These models are evaluated on a predefined test set consisting of 0.1M samples. The test MSE of these models are 0.33, 0.69, and 0.79, respectively, with the initial (w/o training) test MSE of 1.241. Fig. 11 depicts the attacks using these three models on the same agent in Ant(4x2) environment using $\|\cdot\|_\infty$ budget constraint. Interestingly, the dynamics model trained with only 0.2M samples for 1 epoch can achieve comparable performance with the other two models using $5\times$ more samples.

In addition to the visualization in Fig. 1, we investigate how the state values change during these attacks. Fig. 12 presents different recordings of state values under adversarial attacks compared to no attack. Consider state index 8, which represents the horizontal velocity of the agent. For the **HalfCheetah(2x3)** environment, as the goal is to make the agent move forward as fast as possible, we expect the reward to be proportional to this state value. From Fig. 12, all three attacks have fairly sparse red fractions across time-steps, which result in a much lower reward compared to the no-attack setting. Among the three attacks, our model-based ones appear to have the most sparse red fractions leading to the lowest rewards. In addition, the model-based attack appears to show its advantage in environments with more agents as our approach results in lower rewards under a smaller budget as seen in **HalfCheetah(6x1)** and **Ant(4x2)** environments. Additionally, we record the evolution of noise values and the results are shown in Fig. 13.

In summary, our c-MBA attack is able to shows its advantage in all tested multi-agent environment where it achieves lower reward with smaller budget level. Moreover, c-MBA with the victim agent selection has been shown to constantly performs better than the original c-MBA variant as seen in Figure 3.

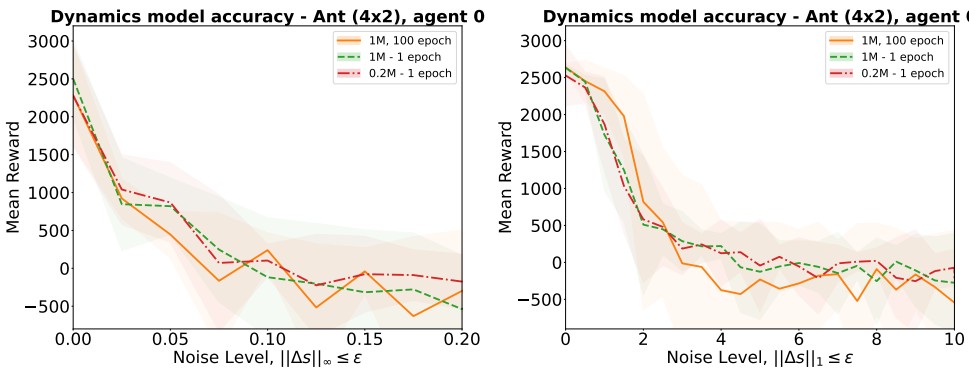

Figure 11: Effect of using three dynamics models in **Ant(4x2)** when attacking Agent 0 out of four agents in **Ant(4x2)** - **Exp. (VI)**.

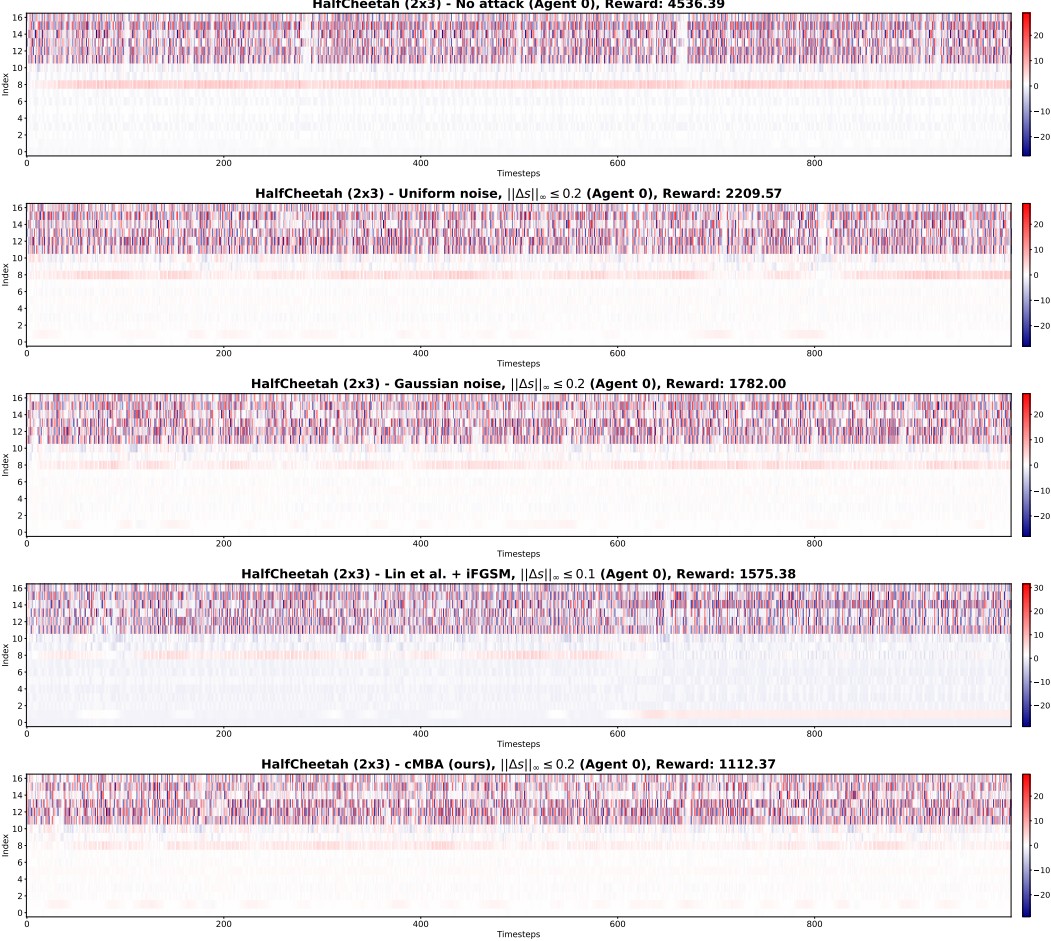

Figure 12: Recordings of state values in an episode under different attacks one out of two agents (Agent 0) in **HalfCheetah( 2x3)** environment.

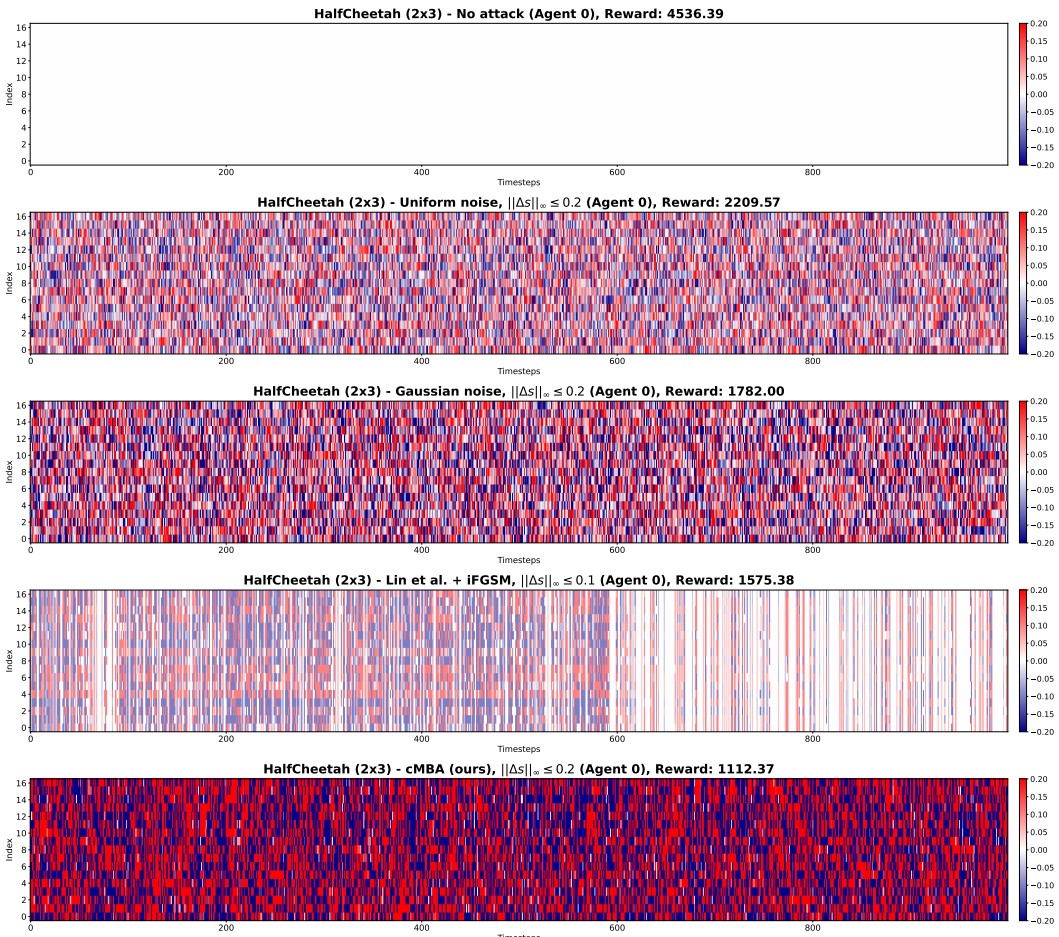

Figure 13: Recordings of noise values in an episode under different attacks on Agent 0 in **HalfCheetah( 2x3)** environment.

