# OpenReview forum: "Evaluating Robustness of Cooperative MARL: A Model-based Approach"
_ICLR.cc/2023/Conference — Submitted to ICLR 2023_

### Official Review · Reviewer_SrTs · 2022-10-22

**Confidence:** 3
**Correctness:** 3
**Technical Novelty And Significance:** 2
**Empirical Novelty And Significance:** 2
**Recommendation:** 3

**Clarity, Quality, Novelty And Reproducibility:**

The clarity of this paper is generally good, but the writing of this paper should be improved. For example, it is better to give a very brief outline before moving to the details in Section 3.

The originality (or novelty) of this work is comparatively minor, since it just combines the techniques in adversarial attacks and model-based RL together. I agree that this is an improvement but not a major breakthrough work, since everything seems unsurprising.

The reproducibility of this work is good, since the authors provide detailed experimental setups and pseudo codes are clear.

Overall, the quality of this work is not good enough.

**Details Of Ethics Concerns:**

This paper studies the methodology of attacking the cooperative multi-agent reinforcement learning algorithms, which could lead to some potentially harmful insights and security issues though they claim that this is for evaluating the robustness of algorithms.

**Strength And Weaknesses:**

## Strength
1. The description of the proposed method is concrete and comparison to the related work is clear.
2. The experimental design is comparatively fair to demonstrate the effectiveness of the proposed method.


## Weaknesses
1. There are multiple editing errors which should be addressed in the modified version.
2. The authors claim that in the setting of Dec-POMDP each agent receive an individual reward. However, as far as I know there should be only one team reward. It should be a factual error.
3. The author also claim that "The requirement on training an adversarial policy is impractical and expensive compared to learning the dynamics model.", which has not been clearly verified. For example, how is the sample efficiency between the off-policy methods and the model-based method?
4. The authors claim that "In our experiments, we notice that the quality of the dynamics model does not significantly affect the result as seen in Figure 11.". To me, it is so wierd. I wonder whether learning a dynamic model is a key factor in this method. For example, I wish the authors can show whether a linear model with randomized parameters that characterizes the relationship between state-action pair and next state can still achieve a comparatively good result.
5. About Eq. (5), if I understand it correctly, the $W_{i}$ obtained is only for ranking the victim-agents and the state perturbations are not penalised with $W_{i}$. In other words, the coefficient prior to each perturbation is either 0 or 1. If it is correct, this is inconsisitent with the formulation in Eq. (4). I wish the authors can give more explanation for it.

**Summary Of The Paper:**

This paper mainly studies a method to evaluate the robustness of c-MARL agents. The main contributions of this paper are using model-based appoach rather than the model-free approach. The most critical benefit of this approach is that the learned dynamic model can be used to generate the next failure state. In addition, the authors also formulate a mixed integer problem to locate the vulnerable agent (which is named as victim-agent in the paper) to inject stronger attacks to damage the whole multi-agent system.

**Summary Of The Review:**

This paper describes a methodology to attack c-MARL with model-based approach, but there are still multiple concerns about the proposed approach which need to be addressed.

---

### Official Review · Reviewer_oW7x · 2022-10-25

**Confidence:** 4
**Correctness:** 3
**Technical Novelty And Significance:** 2
**Empirical Novelty And Significance:** 2
**Recommendation:** 5

**Clarity, Quality, Novelty And Reproducibility:**

__Novelty__

There are many other papers that study attacking (cooperative) multi-agent reinforcement learning. The authors may want to at least discuss these papers in the related work section.

- Guo et al., Adversarial policy learning in two-player competitive games
- Wang et al., Backdoorl: Backdoor attack against competitive reinforcement learning
- Pham et al., Evaluating robustness of cooperative MARL
- Nisioti et al., Robust multi-agent q-learning in cooperative games with adversaries.

**Strength And Weaknesses:**

The main concern of the reviewer is about the novelty and applicability of the proposed method.

- The proposed attacking method depends on a pre-trained dynamics model. Training such a model may be costly in multi-agent settings.

- The most frustrating point about the proposed method is that the target state needs to be collected, which largely limits the applicability of the proposed method to sparse reward settings and tasks with large action-oberversation space.

- The proposed method has access to the policy of all agents. Such a white-box attacking method is less applicable.

**Summary Of The Paper:**

The paper proposes a method that attack cooperative multi-agent reinforcement learning. The major features are
- Model-based: predict future states and find those with low rewards.
- Continuous action space: distinguish the paper from those on discrete action space.

**Summary Of The Review:**

The research direction is interesting and promising, but the proposed method relies on less realistic assumptions and models.

---

### Official Review · Reviewer_giiS · 2022-10-31

**Confidence:** 4
**Correctness:** 3
**Technical Novelty And Significance:** 3
**Empirical Novelty And Significance:** Not applicable
**Recommendation:** 5

**Clarity, Quality, Novelty And Reproducibility:**

The presentation of the paper is clear. The authors also provide the code. The novelty is ok, although this problem has been solved by other paper before, this paper introduce a novel method on victim agent selection, which has never been explored before, to the best of my knowledge.

**Strength And Weaknesses:**

Strong point:
+ This paper tries to design a model-based attacking algorithm for MARL, this problem has not been explored too much before.
+ A novel victim selection algorithm is proposed, which is also novel.
+ Evaluation results shows that the performance is great.

Weak point:
- It seems that MARL is very easy to attack, according to Figure 5 and 6. This is also mentioned in the early work (Lin et al. 2020). The author should justify if their approach is still necessary given its relevant high implementation cost.
- The other problem is training this attack method requires the attacker to have access to the RL environment and model parameters, which may seriously limited the application of this approach.
- Compared with the early work (Lin et al. 2020) The novelty is not that obvious.

**Summary Of The Paper:**

In this paper, the author propose a novel model-based method to attack the team performance in cooperative Multi-agent reinforcement learning. By changing the input state of the victim agent, the goal is to reduce the reward function of the whole team. In addition, the authors also propose a  victim-agent selection strategy to choose the most vulnerable agent to attack. The evaluation results shows that the proposed method can consistently outperforms other baselines in all tested environments.

**Summary Of The Review:**

Overall, I think secure MARL is an interesting and important topic. The paper did bring some novelty to the field -- proposing an algorithm to select the victim agent which is the most vulnerable. However, the author should justify the applicability of their approaches, as mentioned in the Strength And Weaknesses section.

---

### Official Review · Reviewer_Rtpm · 2022-11-04

**Confidence:** 3
**Correctness:** 3
**Technical Novelty And Significance:** 2
**Empirical Novelty And Significance:** 2
**Recommendation:** 5

**Clarity, Quality, Novelty And Reproducibility:**

The paper is well motivated and clearly written.

The research topic of studying the vulnerability of multi-agent cooperative RL methods is interesting. The idea of applying a environmental transition model to find the best state perturbation is novel.

The code is also attached in the Appendix.

**Strength And Weaknesses:**

**Strength**
  * The research topic is interesting and the paper is well-motivated and clearly written.

**Weaknesses**
  * The definition of the failure states in Section 3.2 is too simple. Since the target of RL is to maximize the expected cumulative reward, simply minimizing the immediate reward the agent can obtain is not a good practice and may get in suboptimal solutions. Instead, we should find a target state which could minimize the Q-value, i.e., the expected cumulative reward, of the RL agent.
  * I think there exist more simple but effective baselines. Since the authors perform adversarial attacks on agents trained using MADDPG. The most straightforward way to apply adversarial attacks to MADDPG is:
    * $\min _{\Delta s=\left(\Delta s^1, \cdots, \Delta s^n\right)} Q(\mathbf{s}, \mathbf{\hat{a}})$, s.t., $\mathbf{s}=\text{concat}(s_1, \ldots, s_N)$ and $\mathbf{a}=\text{concat}(\hat{a_1}, \ldots, \hat{a_N})$, where $\hat{a_i}=\pi^i\left(s_i+\Delta s^i\right), \forall i \in \mathcal{N}$ and $Q(\mathbf{s}, \mathbf{\hat{a}})$ is the centralized critic Q-function of MADDPG.
    * In other words, the idea is to modify the input state to the policy network of the agents such that the $Q(\mathbf{s}, \mathbf{\hat{a}})$ value of MADDPG can be minimized. This optimization problem can be implemented via SGD methods. As this method is very simple to implement, I'd like to see how well does this simple algorithm perform compared with the proposed (a little more complicated) one.
  * What's the performance of the proposed method on some more complicated MARL benchmarks, e.g., SMAC?

**Summary Of The Paper:**

This paper proposes a model-based adversarial attack framework (c-MBA) on continuous action spaces of cooperative MARL. The core idea is to find the best perturbations to the states of the agents' policy networks using a transition model such that the state after the transition will be the closest to the failure state. The idea of finding the most vulnerable victim agents is interesting.

**Summary Of The Review:**

This paper proposes a model-based adversarial attack framework method to evaluate the robustness of multi-agent cooperative RL methods. The idea is novel and interesting. But some parts of the method designs can be further improved. The reviewer also thinks that there exist more simple but effective baselines. So, the reviewer currently recommends a weak reject.

---

### Official Review · Reviewer_Hx5n · 2022-11-04

**Confidence:** 3
**Correctness:** 4
**Technical Novelty And Significance:** 3
**Empirical Novelty And Significance:** 3
**Recommendation:** 6

**Clarity, Quality, Novelty And Reproducibility:**

Clarity: The manuscript is generally clear and understandable.

Quality: High.

Novelty: Medium.

Reproducibility: The code is not included as part of the submission.

**Strength And Weaknesses:**

## Strengths

- I think the problem and approach are interesting and novel. A model-based approach for adversarial attacks on c-MARL settings seems promising. Furthermore, leveraging the unique setting in MARL to select the most vulnerable agent to attack is novel and intuitive.
- The authors present numerous experiments to demonstrate the superior performance of their method against simple and model-free baselines for adversarial attacks.

## Weaknesses

- I understand the method is designed for continuous action tasks. I believe it should, with a straightforward change, also work on discrete action settings. If so, it would be beneficial to apply it to SMAC and compare it directly with Lin et al. (2020). I think this would strengthen the arguments made in the paper.
- Applying it to problems would a larger number of agents would also strengthen the paper.

Note: My score is not final and can change (in either direction) based on the authors' responses or the feedback of other reviewers.

**Summary Of The Paper:**

The authors propose a new attack method, called c-MBA, for evaluating the robustness of cooperative multi-agent reinforcement learning (c-MARL) environment. c-MBA algorithm is a model-based attack to craft adversarial observation perturbations. The authors empirically demonstrate c-MBA’s superior performances against model-free baselines in two domains: multi-agent MuJoCo and particle benchmarks.

**Summary Of The Review:**

This is an interesting paper where the authors propose model-based approach for adversarial attacks on c-MARL settings with continuous actions (for the first time). It includes extensive empirical analysis and ablation studies. I think the paper could also use discrete action environments and compare c-MBA directly with other baselines (designed specifically for those settings).

---

### Decision · Program_Chairs · 2023-01-20

**Decision:**

Reject

**Justification For Why Not Higher Score:**

The assumptions of this attack method are very strong: It assumes access to both the RL environment and the victims' policy, which could significantly limit the potential applications of this approach. This is more worrisome in combination with the assumption that the value function (critic) is not available to rule out simple baselines. Additionally, the novelty of this method compared to previous work (Lin et al. 2020) is limited. In particular, using a "target agent" seems like a small change compared to the baselines.
It is also unclear how difficult it would be to defend against these attacks and the paper fails to evaluate attacks in partially observable settings.

**Justification For Why Not Lower Score:**

NA

**Metareview: Summary, Strengths And Weaknesses:**

Summary:
This paper proposes a novel adversarial attack framework for cooperative MARL which uses model based a model based approach to finding more efficient attacks.

Strength:
Adversarial robustness is an important and under-explored topic in cooperative MARL.

Weakness:
The assumptions of this attack method are very strong: It assumes access to both the RL environment and the victims' policy, which could significantly limit the potential applications of this approach. This is more worrisome in combination with the assumption that the value function (critic) is not available to rule out simple baselines. Additionally, the novelty of this method compared to previous work (Lin et al. 2020) is limited. In particular, using a "target agent" seems like a small change compared to the baselines.
It is also unclear how difficult it would be to defend against these attacks and the paper fails to evaluate attacks in partially observable settings.